# $p$-value Adjustment for Monotonous, Unbiased, and Fast Clustering Comparison

**Kai Klede**[1]    **Thomas Altstidl**[1]    **Dario Zanca**[1]    **Björn Eskofier**[1,2]

[1]Machine Learning and Data Analytics (MaD) Lab
Friedrich-Alexander Universität Erlangen-Nürnberg
[2]Translational Digital Health Group
Institute of AI for Health, Helmholtz Zentrum München
`{kai.klede; thomas.r.altstidl; dario.zanca; bjoern.eskofier}@fau.de`

## Abstract

Popular metrics for clustering comparison, like the Adjusted Rand Index and the Adjusted Mutual Information, are type II biased. The Standardized Mutual Information removes this bias but suffers from counterintuitive non-monotonicity and poor computational efficiency. We introduce the $p$-value adjusted Rand Index ($\mathrm{PMI_2}$), the first cluster comparison method that is type II unbiased and provably monotonous. The $\mathrm{PMI_2}$ has fast approximations that outperform the Standardized Mutual information. We demonstrate its unbiased clustering selection, approximation quality, and runtime efficiency on synthetic benchmarks. In experiments on image and social network datasets, we show how the $\mathrm{PMI_2}$ can help practitioners choose better clustering and community detection algorithms.

## 1 Introduction

Clustering is fundamental to unsupervised learning, and practitioners can choose from many algorithms to partition a dataset into homogeneous clusters. Therefore it is common to annotate parts of an otherwise unlabeled dataset and select the clustering algorithm that best reproduces the annotations [1]. A good selection crucially depends on the clustering comparison method, like the Mutual Information (MI) [25] or the Rand Index (RI) [39]. The importance of the comparison method further increases with the advent of deep learning methods for applications such as community detection, in which they serve as components of loss functions during network training [10, 12, 33]. Some use cases admit multiple clustering solutions [23, 38], and clustering comparison can help identify qualitatively different clustering solutions for a single dataset. Other applications of clustering comparison include categorical feature selection or consensus clustering [5, 17].

The MI and RI are biased towards clusterings with particular cluster size distributions [19] (type I bias). For example, the MI favors larger clusters for any reference clustering [24]. The Adjusted Rand Index (ARI) [11] and Adjusted Mutual Information (AMI) [25] achieve a constant baseline value by subtracting the expected value under random permutation of the cluster labels. However, they still exhibit a bias when multiple clusterings are compared via a fixed ground truth (type II bias) [29], as opposed to comparing two random clusterings with each other. This type II scenario typically arises when selecting the best algorithm for a given task on a labeled subset of the data.

Romano et al. [30] showed via the Tsallis entropy, that the AMI and ARI are special cases of generalized information-theoretic clustering comparison measures $\mathrm{AMI}_q$ and proposed standardization to resolve both types of biases. However, the runtime for standardization generally exhibits a substantial time complexity of $\mathcal{O}(N^3 k_A \max(k_A, k_B))$ [30], where $N$ represents the number of data points and $k_A$ and $k_B$ denote the respective number of clusters. This complexity is prohibitive for many applications [15]. Gösgens et al. [9] found that the Standardized Mutual Information (SMI) does not

37th Conference on Neural Information Processing Systems (NeurIPS 2023).

increase monotonically as one clustering is adjusted to match a reference and therefore reject the SMI entirely.

This work presents the $p$-value of the $\mathrm{MI}_q$ (denoted $\mathrm{PMI}_q$) as a provably monotonous type II bias correction. We formally define type II bias and prove that the $\mathrm{PMI}_q$ does not suffer from it. For the $\mathrm{SMI}_q$, there is only empirical evidence [29, 30]. We show that the $\mathrm{PMI}_q$ is monotonous for $q \geq 2$. This includes the $p$-value of the RI, but for the MI, the $p$-value is not monotonous. When normalized with the normal CDF, the $\mathrm{SMI}_q$ approximates the $\mathrm{PMI}_q$, which we confirm via Monte Carlo simulation. We reduce the runtime of the $\mathrm{SMI}_2$ from $\mathcal{O}(N^3 k_A \max(k_A, k_B))$ to a much more practical $\mathcal{O}(k_A k_B)$ by a reformulation of the variance term. We demonstrate the impact of type II unbiased algorithm selection for community detection on a social network dataset and clustering on images of handwritten digits and human faces.

## 2 Generalized information theoretic clustering comparison measures

A clustering $A$ of $N$ data points is a partition of the set $\{1, \ldots, N\}$ into disjoint subsets $A = \{A_1, \ldots, A_{k_A}\}$. $A_i$ denotes the set of points in the $i$-th cluster of size $a_i := |A_i|$ and $k_A$ is the number of clusters in $A$. Clustering comparison measures quantify the similarity between two clusterings $A$ and $B$, and can be expressed as a function of the contingency table (Table 1).

Table 1: Contingency table for clusterings $A, B$ with $k_A$ and $k_B$ clusters respectively. Lower case $a_i$ and $b_j$ denote the cluster size of the $i$-th and $j$-th clusters in $A$ and $B$, while $n_{ij}$ represents the size of their overlap. Clustering comparison measures can be expressed in terms of the elements of this contingency table.

|   |   | $B$ | | | | |
|---|---|---|---|---|---|---|
|   |   | $b_1$ | $\cdots$ | $b_j$ | $\cdots$ | $b_{k_B}$ |
|   | $a_1$ | $n_{11}$ | $\cdots$ | $\cdot$ | $\cdots$ | $n_{1k_B}$ |
|   | $\vdots$ | $\vdots$ | | $\vdots$ | | $\vdots$ |
| $A$ | $a_i$ | $\cdot$ | | $n_{ij}$ | | $\cdot$ |
|   | $\vdots$ | $\vdots$ | | $\vdots$ | | $\vdots$ |
|   | $a_{k_A}$ | $n_{k_A 1}$ | $\cdots$ | $\cdot$ | $\cdots$ | $n_{k_A k_B}$ |

While many clustering comparison methods exist in the literature [2, 9], many well-known methods like the Variation of Information, Mirkin Index, or Rand Index belong to the family of generalized information-theoretic clustering comparison measures [30, 31]. When adjusted for chance, these measures reduce to the mutual information with Tsallis $q$-entropy [35], which will be the focus of this work.

**Definition 2.1** (Tsallis $q$-entropy). Let $q \in \mathbb{R}_+$, and $A$ be a clustering. Then, the *Tsallis $q$-entropy* is

$$H_q(A) = -\sum_{i=1}^{k_A} \left(\frac{a_i}{N}\right)^q \log_q \frac{a_i}{N}, \tag{1}$$

with the $q$-logarithm $\log_q(y) := (y^{1-q} - 1)/(1-q)$ if $q \neq 1$ and the natural logarithm for $q = 1$, where $x \log_1(x) = 0$ for $x = 0$.

The generalized mutual information is defined in analogy to the mutual information but with Tsallis $q$-entropy replacing the Shannon entropy.

**Definition 2.2** (Generalized mutual information). Let $A, B$ be two clusterings of the set $\{1, \ldots, N\}$ and $q \in \mathbb{R}_+$, then the *generalized mutual information* is

$$\mathrm{MI}_q(A, B) = H_q(A) + H_q(B) - H_q(A, B). \tag{2}$$

Here $H_q(A, B)$ denotes the joint $q$-entropy $H_q(\{A_i \cap B_j \mid A_i \in A \wedge B_j \in B\})$.

## 3 Adjustment for chance

The bare $\mathrm{MI}_q$ has limited value as a clustering comparison measure. When comparing two clusterings directly with one another, it is biased towards larger or smaller clusters, depending on the value of

$q$ (type I bias) [19, 25]. But even after adjusting for type I bias, there is another, more subtle bias when multiple clusterings are compared via a single ground truth [29] (Figure 1). In Section 3.2, we introduce the $p$-value as an adjustment to the latter type II bias.

## 3.1 Type I bias

It is well known throughout the literature that the $\mathrm{MI}_1$ is biased towards smaller clusters in direct clustering comparisons [19, 25]. To make this precise, Gösgens et al. [9] defined a family of clustering distributions for which the expected similarity to a reference clustering should be constant:

**Definition 3.1** (Element-symmetric distribution)**.** A distribution over clusterings $\mathcal{B}$ is *element-symmetric* if every two clusterings $B$ and $B'$ with the same cluster sizes have the same probability.

An example of an element-symmetric distribution is the uniform distribution over all clusterings of $N$ elements into $k$ clusters. If the clustering is random, a comparison measure should not favor one particular $k$ over another. If it does, we call it *type I biased* [9].

**Definition 3.2** (Type I unbiased)**.** A clustering measure $V$ is *type I unbiased* if there is a constant $c$, such that for any clustering $A$ with $1 < k_A < N$ and every element-symmetric distribution $\mathcal{B}$ the expected value $\mathbb{E}_{B \sim \mathcal{B}}[V(A, B)] = c$ is constant.

In other words, type I unbiased means that when comparing a fixed clustering $A$ to all permutations of any clustering $B$, the average metric value is the same for all $A$. As the $\mathrm{MI}_q$ has this type I bias, it is commonly adjusted by subtracting its expected value under random permutation, yielding a type I unbiased measure [30].

$$\mathrm{AMI}_q(A, B) := \frac{\mathrm{MI}_q(A, B) - \mathbb{E}_{\sigma \in S_N}[\mathrm{MI}_q(A, \sigma(B))]}{\frac{1}{2}(H_q(A) + H_q(B)) - \mathbb{E}_{\sigma \in S_N}[\mathrm{MI}_q(A, \sigma(B))]}. \tag{3}$$

$S_N$ denotes the symmetric group and $\frac{1}{2}(H_q(A) + H_q(B))$ is an upper bound to the $\mathrm{MI}_q$ such that the $\mathrm{AMI}_q$ is normalized to $c = 0$ for random clusterings and upper bounded by 1 [25, 30]. Adjustments with respect to other random models are possible [8, 18]. However, the random permutation model remains the most popular and is the focus of this work. Due to the generalization using Tsallis entropy, the $\mathrm{AMI}_2$ corresponds to the Adjusted Rand Index [11, 30].

## 3.2 Type II bias

However, in a typical external validation scenario, a single absolute value of the $\mathrm{AMI}_q$ is of little help. While it is easy to understand that an $\mathrm{AMI}_q$ of zero means a clustering algorithm is no better than random and a value of one means optimal agreement, the scale of the range in between is unclear. Therefore, the $\mathrm{AMI}_q$ values of multiple candidate solutions with a reference are typically compared against each other to find the best algorithm for a given dataset.

As a toy model for this scenario, we uniformly generate 5000 clusterings of $N = 500$ elements for each number of clusters $k_B \in \{2, 6, 10, 14, 18, 22\}$ [29]. We compare them to a fixed clustering $A$ with $k_A = 10$ evenly sized clusters and plot the selection probabilities for the RI, MI, and $\mathrm{AMI}_q$ for $q \in \{1, 2\}$ (Figures 1a, b, d and e). The bare MI and RI and their adjusted variants $\mathrm{AMI}_q$ are biased towards certain values of $k_B$.

We generalize and formalize this observation by demanding a clustering comparison measure to favor no element-symmetric distribution over another.

**Definition 3.3** (Type II unbiased)**.** Let $V$ be a clustering comparison measure and $\mathcal{B}, \mathcal{B}'$ be element-symmetric clustering distributions. $V$ is *type II unbiased* if

$$\mathbb{E}_{B \sim \mathcal{B}, B' \sim \mathcal{B}'}[\theta(V(A, B) - V(A, B'))] = \frac{1}{2} \tag{4}$$

for any clustering $A$ with $1 < k_A < N$, where $\theta$ denotes the Heaviside step function with $\theta(0) = 1/2$.

Intuitively, Type I bias means that certain cluster sizes receive higher metric values. Type II bias, on the other hand, gives a higher relative rank to certain cluster sizes when multiple clusterings are compared with a ground truth.

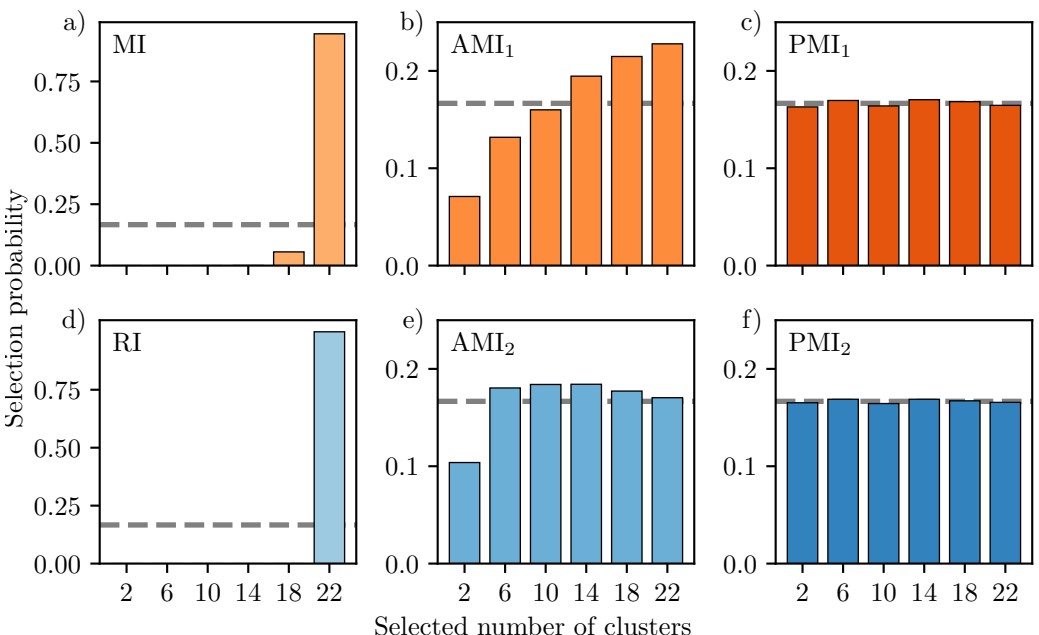

Figure 1: We compare a fixed reference clustering with $k_A = 10$ even clusters, to random clusterings with $k_B \in \{2, 6, 10, 14, 18, 22\}$ clusters. The plot shows the selection probabilities of each $k_B$ for the MI and RI and its adjusted ($\text{AMI}_q$) and our $p$-value adjusted ($\text{PMI}_q$) variants after 5000 repetitions. The RI, MI, and $\text{AMI}_q$ are type II biased, while our $\text{PMI}_q$ selects each cluster size with equal probability.

Romano et al. [29] introduced standardization to correct for type II bias

$$\text{SMI}_q(A, B) := \frac{\text{MI}_q(A, B) - \mathbb{E}_{\sigma \in S_N}[\text{MI}_q(A, \sigma(B))]}{\sqrt{\mathbb{E}_{\sigma \in S_N}[\text{MI}_q(A, \sigma(B))]^2 - \mathbb{E}_{\sigma \in S_N}[\text{MI}_q(A, \sigma(B))^2]}}. \tag{5}$$

They observed in numerical simulations that in the toy model above (Figure 1), the $\text{SMI}_q$ selects each $k_B$ with approximately equal probability [30].

However, the $\text{MI}_q$ is not normally distributed under random permutation (Figure 2a), and standardization is only an approximation to the true $p$-value (Figure 2b). The $p$-value quantifies what percentage of all permutations of the data would have led to higher mutual information, and we propose to use it for clustering comparison.

**Definition 3.4** ($p$-value adjusted, generalized mutual information)**.** Let $A, B$ be two partitions of the set $\{1, \dots, N\}$, $q \in \mathbb{R}_+$. Assuming the random permutation model, the $p$-value adjusted, generalized mutual information is

$$\text{PMI}_q := \mathbb{E}_{\sigma \in S_N}[\theta(\text{MI}_q(A, B) - \text{MI}_q(\sigma(A), B))]. \tag{6}$$

Note that as the marginal entropies are independent of the permutation, $\text{PMI}_q = \mathbb{E}_{\sigma \in S_N}[\theta(H_q(\sigma(A), B) - H_q(A, B))]$. For $q = 1$, this is the $p$-value of the mutual information and the variation of information by definition. For $q \neq 1$ the $\text{PMI}_q$ further simplifies to $\mathbb{E}_{\sigma \in S_N}[\theta(n'^q_{ij} - n^q_{ij})]$ for $q > 1$ and $\mathbb{E}_{\sigma \in S_N}[\theta(n^q_{ij} - n'^q_{ij})]$ for $q < 1$ with $n'_{ij}$ being the elements of the contingency table for $\sigma(A), B$. In a sense, the details of the generalized mutual information don't matter under $p$-value adjustment, except the exponent $q$ of the contingency matrix elements. In fact, the $p$-value of the Rand Index is equivalent to the $\text{PMI}_2$ by a very similar argument.

In the experiment in Figure 1, we observe that the $\text{PMI}_1$ and $\text{PMI}_2$ select each $k_B$ with approximately equal probability.

**Proposition 3.1.** The $\text{PMI}_q$ is type I and type II unbiased.

We formally prove this result in Appendix A.

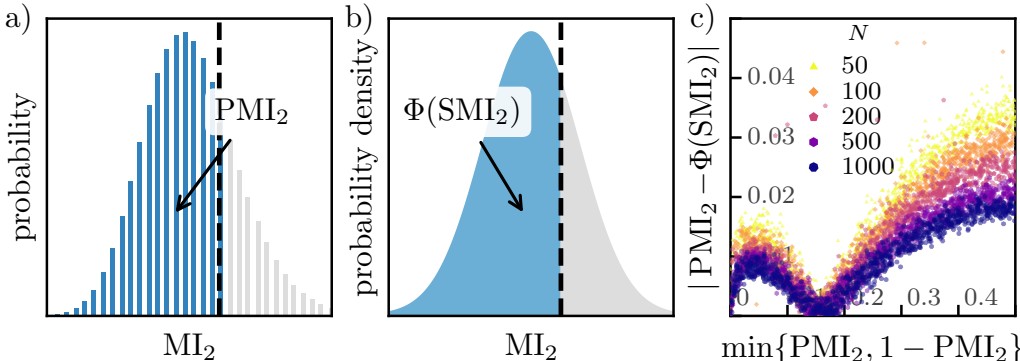

Figure 2: The probability of obtaining a particular $\mathrm{MI}_2$ under random permutation for two fixed clusterings $A, B$ each of 100 elements. Our $\mathrm{PMI}_2$ (blue bars in a)) takes the true distribution of the $\mathrm{MI}_2$ into account, whereas the $\mathrm{SMI}_2$ (shaded blue region in b)) is based on a continuous normal approximation. However, when normalized with the normal CDF $\Phi$, the $\mathrm{SMI}_2$ is a good approximation of the $\mathrm{PMI}_2$ as shown in Figure c). Here, we sampled 1000 pairs of clusterings uniformly at random for different numbers of elements $N$. We plot the absolute difference between Monte Carlo estimates of the $\mathrm{PMI}_2$ and normalized $\mathrm{SMI}_2$ values as a function of the two-sided $p$-value. The larger the dataset size $N$, the better $\Phi(\mathrm{SMI}_2)$ approximates the true $\mathrm{PMI}_2$.

## 4 Monotonicity

When one clustering is changed to resemble another more, any clustering similarity measure should increase. A major drawback of the $\mathrm{SMI}_1(A, B)$ is its non-monotonous behavior as the number of pairs of elements that agree in $A$ and $B$ increases [9]. We show that the $\mathrm{PMI}_q$, on the other hand, is monotonous for $q \geq 2$.

### 4.1 Definition of monotonicity

The atomic operations that add new pairs of agreeing elements in two clusterings are the *perfect split* and the *perfect merge* [9].

**Definition 4.1** (Perfect split). $B'$ is a perfect split of $B$ with respect to $A$ if it splits a single cluster $B_1 \in B$ into $B_1', B_2' \in B'$ such that for all $i$ either $A_i \cap B_1 \subset B_1'$ or $A_i \cap B_1 \subset B_2'$.

**Definition 4.2** (Perfect merge). $B'$ is a perfect merge of $B$ with respect to $A$ if $B'$ is obtained by merging two clusters $B_1, B_2 \in B$ with $B_1, B_2 \subset A_i$ for some $i$.

Gösgens et al. [9] require that any clustering similarity measure increases monotonically for any combination of perfect splits and perfect merges.

**Definition 4.3** (A-consistent improvement). $B'$ is an *A-consistent improvement* of $B$ iff there exists a series of perfect splits and perfect merges that change $B$ into $B'$.

**Definition 4.4** (Monotonicity). A symmetric clustering comparison measure $V$ is monotonous if for every $A, B$ with $1 < k_A < N$ and any $A$-consistent improvement $B'$ of $B$, $V(A, B') > V(A, B)$.

We show that the $\mathrm{PMI}_q$ is monotonous for $q \geq 2$, making it the first known clustering comparison measure to be both type II unbiased and monotonous. The case $q = 2$ is particularly interesting as it corresponds to the well-known Rand Index.

### 4.2 Proof of monotonicity for $\mathrm{PMI}_q$ with $q \geq 2$

The proof can be broken down into monotonicity under perfect splits and perfect merges. We first show that the joint $q$-entropy increases under any split that is not perfect.

**Lemma 4.1.** Let $A, B$ be clusterings with $1 < k_A < n$ and $B'$ be obtained by splitting a cluster $B_j \in B$ into non-empty clusters $B'_{j_1}, B'_{j_2}$. Then $H_q(A, B') = H_q(A, B) + \Delta H_q$ with $\Delta H_q \geq 0$ and equality iff the split is perfect with respect to $A$.

This statement is a direct consequence of the subadditivity of the $q$-entropy [7]. In particular $h_q : p \mapsto p^q \log_q p$ is a strictly convex function with $h_q(0) = 0$ and hence strictly superadditive, i.e. $h_q(p_1 + p_2) \geq h_q(p_1) + h_q(p_2)$ for any $p_1, p_2 \geq 0$ with equality iff $p_1 = 0 \vee p_2 = 0$.

*Proof of Lemma 4.1.* We express $\Delta H_q$ as

$$\Delta H_q = \sum_i \left[ \left(\frac{n_{ij}}{N}\right)^q \log_q \left(\frac{n_{ij}}{N}\right) - \left(\frac{n'_{ij_1}}{N}\right)^q \log_q \left(\frac{n'_{ij_1}}{N}\right) - \left(\frac{n'_{ij_2}}{N}\right)^q \log_q \left(\frac{n'_{ij_2}}{N}\right) \right]$$

$$= \sum_i h_q \left(\frac{n_{ij}}{N}\right) - h_q \left(\frac{n'_{ij_1}}{N}\right) - h_q \left(\frac{n'_{ij_2}}{N}\right). \tag{7}$$

From $n_{ij} = n'_{ij_1} + n'_{ij_2}$ and the strict superadditivity of $h_q$ follows $\Delta H_q \geq 0$ with equality iff $n'_{ij_1} = 0 \vee n'_{ij_2} = 0$, i.e. when the split is perfect. $\qquad\square$

Conversely, a perfect merge maximizes the difference in joint entropy.

**Lemma 4.2.** Let $A, B, B'$ and $\Delta H_q$ be as in Lemma 4.1, then for $q \geq 2$, $\Delta H_q$ is maximal iff $B$ is a perfect merge of $B'$ with respect to $A$.

*Proof of Lemma 4.2.* When $B$ is a perfect merge of $B'$ with respect to $A$, then

$$\Delta H_q^{\text{perfect}} = h_q \left(\frac{b_j}{N}\right) - h_q \left(\frac{b'_{j_1}}{N}\right) - h_q \left(\frac{b_j - b'_{j_1}}{N}\right). \tag{8}$$

To show that $\Delta H_q^{\text{perfect}}$ is superadditive as a function of $b_j$, we take its second derivative

$$\frac{\mathrm{d}^2}{\mathrm{d}b_j^2} \Delta H_q^{\text{perfect}} = \frac{q}{N^q} \left( b_j^{q-2} - (b_j - b'_{j_1})^{q-2} \right) \geq 0 \text{ for } q \geq 2 \text{ and } b_j > b'_{j_1} > 0. \tag{9}$$

For $q > 2$, it is strictly convex and thus strictly superadditive. For $q = 2$ and $b_j = 0$, the difference $\Delta H_q^{\text{perfect}} = -b_{j_1}'^2$ is negative and thus $\Delta H_2^{\text{perfect}}$ is also strictly superadditive. Now consider $\tilde{A}$ such that $B$ is not a perfect merge with respect to $\tilde{A}$. Then at least two $\tilde{A}_{i_1}, \tilde{A}_{i_2}$ have non-vanishing overlap $\tilde{n}_{i_1 j}, \tilde{n}_{i_2 j} > 0$ with $B_j$ such that

$$\Delta H_q^{\text{perfect}} > \sum_i h_q \left(\frac{\tilde{n}_{ij}}{N}\right) - h_q \left(\frac{b'_{j_1}}{N}\right) - h_q \left(\frac{\tilde{n}_{ij} - b'_{j_1}}{N}\right) \text{ for } q \geq 2. \tag{10}$$

With the superadditivity of $h_q$ follows $H_q^{\text{perfect}} > H_q^{\text{not perfect}}$ (Compare Eq. 7). $\qquad\square$

Now that we know how the joint entropy behaves under perfect splits and perfect merges, we can put together the proof of the monotonicity of the $\text{PMI}_q$.

**Theorem 4.3.** Let $A, B$ be clusterings with $1 < k_A < n$ and $B'$ an $A$-consistent improvement of $B$. Then $\text{PMI}_q(A, B') > \text{PMI}_q(A, B)$ for $q \geq 2$.

*Proof of Theorem 4.3.* It suffices to show monotonicity for $B'$ a perfect split or perfect merge since any $A$-consistent improvement of $B$ can be obtained by a sequence of perfect splits and perfect merges.

**Case 1.** $B'$ is a perfect split.
Since $A$ is not a singleton cluster, a permutation $\sigma$ exists such that $B'$ is not a perfect split with respect to $\sigma(A)$ and with Lemma 4.1 it follows

$$\text{PMI}_q(A, B') > \mathbb{E}_{\sigma \in S_N} [\theta(H_q(\sigma(A), B) - H_q(A, B'))]. \tag{11}$$

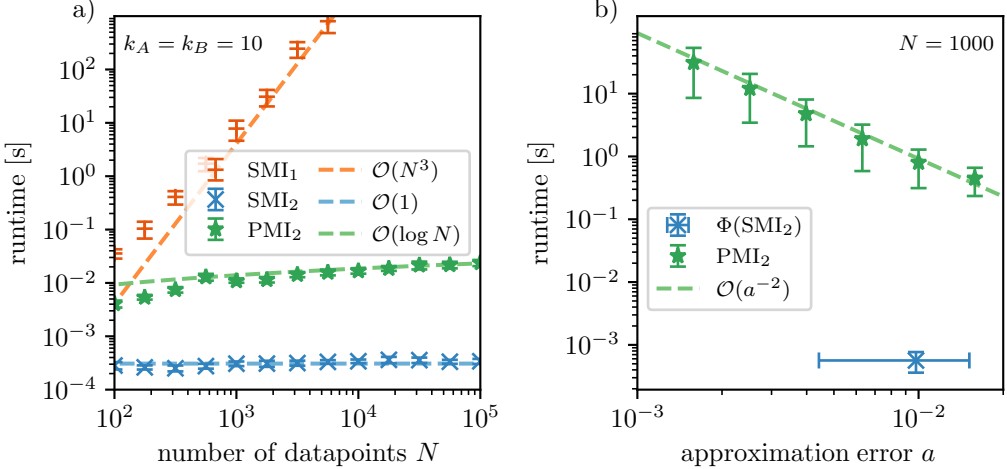

Figure 3: Runtime of the Monte Carlo $\text{PMI}_2$ and the $\text{SMI}_2$ for random clusterings of a) $N$ elements into $k_A = k_B = 10$ clusters and of b) random clusterings with $N = 1000$ and varying approximation error $a$. The $\text{SMI}_1$ calculation, as proposed in [29], is prohibitively expensive for medium-sized datasets. Our exact reformulation of the $\text{SMI}_2$ and the Monte Carlo $\text{PMI}_2$ maintain practical runtimes for high $N$. The $\Phi(\text{SMI}_2)$ is faster, while the Monte Carlo $\text{PMI}_2$ allows for higher accuracy.

However, $B'$ is a perfect split of $B$ with respect to $A$ and equality holds in Lemma 4.1

$$\text{PMI}_q(A, B') > \mathbb{E}_{\sigma \in S_N}[\theta(H_q(\sigma(A), B) - H_q(A, B))] = \text{PMI}_q(A, B). \tag{12}$$

**Case 2.** $B'$ is a perfect merge.
Let $b_1, b_2 \in B$ denote the merged clusters that form $b'_1 \in B'$. Using Lemma 4.2, we find

$$\text{PMI}_q(A, B') = \mathbb{E}_{\sigma \in S_N}[\theta(H_q(\sigma(A), B) - H_q(A, B) + \Delta H_q^{\sigma(A)} - \Delta H_q^A)] > \text{PMI}_q(A, B), \tag{13}$$

as there is at least one permutation $\sigma$ for which the merge is not perfect.

$\square$

## 5 Approximations and runtime

A limitation of the $\text{PMI}_q$ is its computational complexity. Its exact calculation is intractable even for small datasets, as it requires a sum over all contingency tables with given marginals. To mitigate this limitation, we propose two approximation schemes:

1. **Standardized approximation (q = 2):** We approximate the true, discrete distribution of $\text{MI}_q$ with a continuous normal distribution that matches its first and second statistical moments (See Figure 2a and b). While this approximation is particularly fast for $q = 2$, it does not preserve the theoretical guarantees of the $\text{PMI}_q$.

2. **Monte Carlo approximation:** Given two clusterings $A$ and $B$, we sample contingency tables with the same cluster sizes. The fraction of tables with $\text{MI}_q$ lower than $\text{MI}_q(A, B)$ approximates the true $p$-value. In this approach, the theoretical guarantees hold up to a tunable approximation error at the cost of higher runtime.

### 5.1 The standardized Rand Index

We approximate the $\text{PMI}_q$ with the $\text{SMI}_q$, normalized with the normal CDF $\Phi$ (Figure 2b). This can be seen as a truncated, second-order Gram Charlier A series of the $\text{PMI}_q$, and while this could be continued for higher statistical moments, it is difficult to find an exact error term [37]. A more cautious

normalization permits the lower bound $\text{PMI}_q(A, B) \geq 1 - \frac{1}{1 + (\text{SMI}_q(A,B))^2}$ for $\text{SMI}_q(A, B) > 0$, but has little practical significance in the context of this work [30]. Therefore, we evaluate the approximation quality experimentally[1] on 1000 pairs of clusterings drawn uniformly from the set of all clusterings with $N \in \{50, 100, 200, 500, 1000\}$ using a method described in [16]. We compare $\Phi(\text{SMI}_2) := (1 + \text{erf}\,(\text{SMI}_2\,/\sqrt{2}))/2$ with a Monte Carlo estimate of the $\text{PMI}_2$ with approximation error 0.001 in Figure 2c. The values are highly correlated ($r_{\text{Pearson}} = 0.9983$ for $N = 50$), and the approximation improves with larger values of $N$ ($r_{\text{Pearson}} = 0.9995$ for $N = 1000$). So although $\Phi(\text{SMI}_q)$ itself is not monotonous, it closely matches the $\text{PMI}_q$, which is monotonous for $q \geq 2$.

While $\Phi(\text{SMI}_q)$ is a simplification over the $\text{PMI}_q$, its computational complexity $\mathcal{O}(N^3 k_A \max(k_A, k_B))$ for general $q$ is far from practical [15, 30]. In this work, we contribute a novel algorithm for the special case $q = 2$ that improves the computational complexity.

**Proposition 5.1.** The computational complexity of $\text{SMI}_2$ is $\mathcal{O}(k_A k_B)$.

The proof is in Appendix C. This special case $q = 2$ is of particular interest because of the correspondence with the well-known Rand Index and the monotonicity of $\text{PMI}_2$. Our improved algorithm for the $\text{SMI}_2$ allows comparisons of moderately sized clusterings $N \approx 10{,}000$ that are computationally out of reach for, e.g., the $\text{SMI}_1$ (Figure 3a).

## 5.2 Monte Carlo approximation

The standardized approximation has two limitations:

- It is computationally inefficient only for $q \neq 2$.
- There is no guarantee that it preserves the desirable theoretical properties of the $\text{PMI}_2$.

We address both of these limitations by introducing a Monte Carlo approximation at the cost of increased runtime. For two clusterings $A, B$, the method samples contingency tables uniformly from all tables with their respective cluster sizes $a_1, \ldots, a_{k_A}; b_1, \ldots, b_{k_B}$ (Compare Table 1), using the algorithms proposed in [4, 26]. The fraction of samples with $\text{MI}_q$ lower than $\text{MI}_q(A, B)$ is an unbiased estimator of the $\text{PMI}_q$. The sampling procedure terminates when a given approximation error $a$ is reached. This way, the theoretical properties of the $\text{PMI}_q$ are preserved up to the tunable approximation error.

However, lower approximation errors require more samples:

**Proposition 5.2.** The computational complexity of the Monte Carlo $\text{PMI}_q$ is $\mathcal{O}(\min(N, k_A k_B \log N)/a^2)$, with the desired approximation error $a$.

The proof is in Appendix B, and Figure 3 shows an experimental study of the runtime compared to the standardized approximation. The Monte Carlo approach is computationally more expensive, especially for larger datasets. Therefore, the standardized approach is for choice when $q = 2$ and moderate approximation quality is acceptable. The Monte Carlo method should be used if $q \neq 2$ or theoretical guarantees are required.

# 6 Algorithm selection on real-world datasets

## 6.1 $k$-means clustering on image datasets

As a first example, we mimic the synthetic experiment in Figure 1. For several numbers of clusters $k$, we apply $k$-means clustering [21] with 1000 different random seeds. We select the clustering with the highest RI, $\text{AMI}_2$, and $\text{PMI}_2$, approximated by $\Phi(\text{SMI}_2)$ and denote the corresponding number of clusters $k_{\text{selected}}$. Figure 4a shows the selection probabilities of $k_{\text{selected}}$ when compared with a ground truth with $k_{\text{true}}$ clusters for a handwritten digit dataset [6] and Figure 4b for a dataset of human faces [27]. Naturally, all measures favor solutions where $k_{\text{selected}} > k_{\text{true}}$, as a higher number of clusters increases the chances of $k$-means matching the reference decision boundaries. However, the RI and $\text{AMI}_2$ additionally suffer from type II bias, leading to a higher overestimation of $k_{\text{selected}}$

---

[1]The experiments were executed on an AMD Ryzen 9 5950X. The code for all experiments is available at `https://github.com/mad-lab-fau/pmi-experiments`.

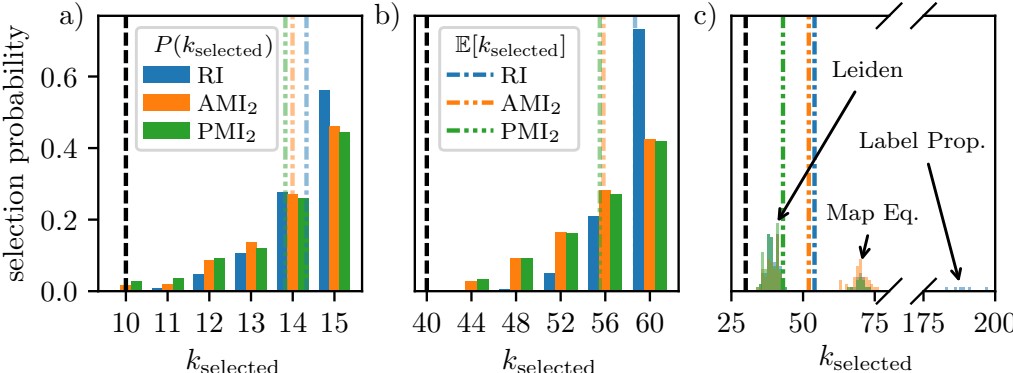

Figure 4: We apply $k$-means clustering [21] with varying $k$ to a) the UCI handwritten digit dataset [6] and b) the Olivetti faces dataset [27] and select the solution with the highest similarity to the ground truth. We repeat this experiment 1000 times with different random seeds and plot the selection probability under the RI, $\mathrm{AMI}_2$, and normal approximation of the $\mathrm{PMI}_2$. The $\mathrm{PMI}_2$ selects candidates where $k_{\mathrm{selected}}$ is closer to the true number of clusters $k_{\mathrm{true}}$ on average (dashed lines) compared to the RI and $\mathrm{AMI}_2$. In c), we select a connected subset of $k_{\mathrm{true}} = 30$ communities from the email EU core dataset [20] and detect communities using five algorithms with 22 parameter configurations. The RI, $\mathrm{AMI}_2$, and $\mathrm{PMI}_2$ select the best solution, and we plot the selection probability for $k_{\mathrm{selected}}$ after 100 repetitions. The $\mathrm{PMI}_2$ prefers the Leiden algorithm, which produces $k_{\mathrm{selected}}$ on the order of $k_{\mathrm{true}}$. The $\mathrm{AMI}_2$ gives a higher probability for Louvain Map Equation, and the RI sometimes selects low-quality Label Propagation results.

(compare with Figure 1). The difference between the $\mathrm{AMI}_2$ and the $\mathrm{PMI}_2$ is subtle, but this is expected. Type II bias correction is just one step forward in clustering comparison and does not turn existing assessments based on metrics like the $\mathrm{AMI}_2$ on its head. In practice, much wider ranges of $k_{\mathrm{selected}}$ can arise from different clustering algorithms which could potentially amplify the effect. Two additional experiments with spectral clustering instead of $k$-means can be found in Appendix E.

## 6.2 Community detection in social networks

In social networks, detecting communities is ubiquitous and can help to detect fraud, deliver personalized content, or target ads [13]. As a result, many community detection algorithms are known in the literature [3, 28, 34]. However, community detection is inherently unsupervised, and it is a priori unclear which algorithm with which parameters will perform best for a given application. In practice, human experts often annotate a subset of the dataset, and an unsupervised algorithm is selected via a clustering comparison measure on that subset.

We simulate this procedure on a network of email conversations between European research institutions, where each institution is a ground truth cluster [20]. We select a connected subset with $k_{\mathrm{true}} = 30$ institutions and detect communities using Degree Ordered Label Propagation, Label Propagation, Leiden, Louvain, and Louvain Map Equation [32] with 22 parameter configurations (Appendix D). We then select the most similar algorithm to the ground truth using RI, $\mathrm{AMI}_2$, and $\mathrm{PMI}_2 \approx \Phi(\mathrm{SMI}_2)$. This process is repeated for 100 subsets per dataset, and the resulting probabilities are shown in Figure 4c. Label propagation is a fast but inaccurate method [28] and overestimates $k_{\mathrm{true}}$ by almost an order of magnitude in our experiment. During algorithm selection, RI was the only metric to choose Label Propagation in some cases. The $\mathrm{PMI}_2$ differs from the $\mathrm{AMI}_2$ in that it selected Leiden more frequently over the Louvain Map Equation, both of which are improvements over the original Louvain method [14, 34]. However, Leiden comes closer to the true number of clusters $k_{\mathrm{true}}$, and in that sense, $\mathrm{PMI}_2$ led to a better choice of algorithm.

Table 2: Comparison of the $\mathrm{PMI}_q$ to the clustering comparison metrics in the systematic review by Gösgens et al. [9]. Examples for type II biasedness can be found in Appendix A. We consider a metric computationally tractable if its asymptotic complexity is linear in the number of data points $N$ but not necessarily in the numbers of clusters $k_A, k_B$. The rationale is that in many cases, the number of clusters is much lower than the number of data points and metrics like the $\mathrm{AMI}_1$ with $\mathcal{O}(N \max\{k_A, k_B\})$ are widely used in practice [27, 29]. The $\mathrm{PMI}_2$ is the first metric to be Type II unbiased and monotonous and, while computationally demanding, has efficient approximations.

| | NMI | $\mathrm{NMI}_{max}$ | Fair NMI | VI | FMeasure | BCubed | Jaccard | Wallace | Dice | Corr. Coeff. | Sokal&Sneath | Corr. Dist. | Rand Index | $\mathrm{AMI}_1$ | $\mathrm{AMI}_2$ | $\mathrm{SMI}_1$ | $\mathrm{SMI}_2$ | $\mathrm{PMI}_1$ | $\mathrm{PMI}_2$ |
|---|---|---|---|---|---|---|---|---|---|---|---|---|---|---|---|---|---|---|---|
| **Type I unbiased** | ✗ | ✗ | ✗ | ✗ | ✗ | ✗ | ✗ | ✗ | ✗ | ✓ | ✓ | ✗ | ✗ | ✓ | ✓ | ✓ | ✓ | ✓ | ✓ |
| **Type II unbiased** | ✗ | ✗ | ✗ | ✗ | ✗ | ✗ | ✗ | ✗ | ✗ | ✗ | ✗ | ✗ | ✗ | ✗ | ✗ | ○ | ○ | ✓ | ✓ |
| **Monotonicity** | ✓ | ✗ | ✗ | ✓ | ✗ | ✓ | ✓ | ✗ | ✓ | ✓ | ✓ | ✓ | ✓ | ✓ | ✓ | ✗ | ✗ | ✗ | ✓ |
| **Comp. tractable** | ✓ | ✓ | ✓ | ✓ | ✓ | ✓ | ✓ | ✓ | ✓ | ✓ | ✓ | ✓ | ✓ | ✓ | ✓ | ✗ | ✓ | ○ | ○ |

# 7 Conclusion and outlook

Table 2 summarizes our findings for the $\mathrm{PMI}_q$ and compares its theoretical properties to 17 clustering comparison measures from the systematic review by Gösgens et al. [9]. We introduce the first type II unbiased and monotonous cluster comparison method, the $p$-value adjusted Rand Index ($\mathrm{PMI}_2$). Existing methods that addressed type II bias, namely the Standardized Mutual Information ($\mathrm{SMI}_1$) and the Standardized Rand Index ($\mathrm{SMI}_2$) are not monotonous, meaning clusterings closer to the ground truth can score worse. In addition, the $\mathrm{SMI}_1$ has high computational complexity, making it unsuitable in practice. For the $\mathrm{SMI}_2$ we showed that an efficient algorithm exists and we leverage this algorithm for an efficient approximation of the proposed $\mathrm{PMI}_2$. However, our analysis of the errors in this standardized approximation is limited to experimental observations, leaving a theoretical analysis for future work. We devised a Monte Carlo approximation for the $\mathrm{PMI}_2$ with tunable approximation error, for when theoretical guarantees are required. To validate our theoretical findings, synthetic experiments confirm that the presented $\mathrm{PMI}_2$ selects different cluster sizes with equal probability and is not subject to type II bias. In practice, the $\mathrm{PMI}_2$ chooses better clustering algorithms from a set of candidates when a ground truth reference is available. Thanks to its monotonicity and computational efficiency, the $\mathrm{PMI}_2$ is a practical candidate for evaluating cluster similarity without type II bias. While we investigated $p$-value adjustment for the family of generalized information-theoretic clustering comparison measures, further research is required to understand if other comparison measures, like the Jaccard Index, could benefit from a similar adjustment.

## Acknowledgments and Disclosure of Funding

This work was funded by the Digital Europe Grant *Testing and Experimentation Facility for Health AI and Robotics (TEF-Health)*, Project number 101100700 and the *Bayerischen Verbundförderprogramm (BayVFP) – Förderlinie Digitalisierung – Förderbereich Informations- und Kommunikationstechnik* of the Bavarian Ministry of Economic Affairs, Regional Development and Energy and supported by *Bayern Innovativ – Bayerische Gesellschaft für Innovation und Wissenstransfer mbH*.

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

# Appendix

## A   Type II Bias

We show that the $\mathrm{PMI}_q$ is both type I and type II unbiased:

*Proof of Proposition 3.1.* The choice of $\theta(0) = \frac{1}{2}$ allows us to use $\theta(x) = 1 - \theta(-x)$ such that

$$
\begin{aligned}
\mathrm{PMI}_q(A, B) = \quad & \mathbb{E}_{\sigma \in S_N}[\theta(H_q(A, B) - H_q(\sigma(A), B))] = \\
& \mathbb{E}_{\tilde{\sigma} \in S_N}[\theta(H_q(\tilde{\sigma}(A), \tilde{B}) - H_q(A, \tilde{B}))] = \\
& 1 - \mathbb{E}_{\tilde{\sigma} \in S_N}[\theta(H_q(A, \tilde{B}) - H_q(\tilde{\sigma}(A), \tilde{B}))],
\end{aligned}
\tag{14}
$$

with $\tilde{B} = \sigma^{-1}(B)$ and $\tilde{\sigma} = \sigma^{-1}$. Now in the definition of type I unbiasedness, $\mathcal{B}$ is element-symmetric, such that $\tilde{B} \in \mathcal{B}$ and

$$
\mathbb{E}_{B \sim \mathcal{B}}[\mathrm{PMI}_q(A, B)] = 1 - \mathbb{E}_{\tilde{B} \sim \mathcal{B}}[\mathrm{PMI}_q(A, \tilde{B})] = \frac{1}{2}.
\tag{15}
$$

Hence, $\mathrm{PMI}_q$ is type I unbiased.

For type II unbiasedness, $\mathcal{B}, \mathcal{B}'$ are element symmetric and with Eq. 14 we get

$$
\begin{aligned}
& \mathbb{E}_{B \sim \mathcal{B}, B' \sim \mathcal{B}'}[\theta(\mathrm{PMI}_q(A, B) - \mathrm{PMI}_q(A, B'))] = \\
& 1 - \mathbb{E}_{B \sim \mathcal{B}, B' \sim \mathcal{B}'}[\theta(\mathrm{PMI}_q(A, B') - \mathrm{PMI}_q(A, B))] = \\
& 1 - \mathbb{E}_{\tilde{B} \sim \mathcal{B}, \tilde{B}' \sim \mathcal{B}'}[\theta(\mathrm{PMI}_q(A, \tilde{B}) - \mathrm{PMI}_q(A, \tilde{B}'))] = \frac{1}{2}.
\end{aligned}
\tag{16}
$$

$\square$

For all other clustering comparison measures in Table 2, the proof or a counterexample to type I unbiasedness are given in [9]. Here, we provide an example for their type II bias. For $N = 4$ data points, we compare the clustering $A = \{\{1, 4\}, \{2, 3\}\}$ with all clusterings with 2 clusters, distributed uniformly $\mathcal{B}$. We also compare $A$ to all clusterings from the uniform distribution $\mathcal{B}'$ of $N = 4$ data points into 3 clusters. All comparison measures except the $\mathrm{PMI}_2$ prefer either 2 or 3 clusters in the sense that their expectation value in Eq. 4 is not equal to $1/2$, and are thus type II biased (See Table 3).

Table 3: Example for type II bias of other clustering comparison measures. For $A = \{\{1, 4\}, \{2, 3\}\}$ and $\mathcal{B}, \mathcal{B}'$ the uniform distribution of all clusterings with 4 elements into 2, 3 clusters respectively, the expected value $\mathbb{E}_{B \sim \mathcal{B}, B' \sim \mathcal{B}'}[\theta(V(A, B) - V(A, B'))] \neq \frac{1}{2}$ and hence these measures are type II biased.

| NMI | $\mathrm{NMI}_{max}$ | Fair NMI | VI | FMeasure | BCubed | Jaccard | Wallace | Dice | Corr. Coeff. | Sokal&Sneath | Corr. Dist. | Rand Index | $\mathrm{AMI}_1$ | $\mathrm{AMI}_2$ | $\mathrm{SMI}_1$ | $\mathrm{SMI}_2$ |
|---|---|---|---|---|---|---|---|---|---|---|---|---|---|---|---|---|
| $\frac{1}{7}$ | $\frac{1}{7}$ | $\frac{11}{21}$ | $\frac{10}{21}$ | $\frac{11}{21}$ | $\frac{13}{21}$ | $\frac{13}{21}$ | $\frac{5}{7}$ | $\frac{13}{21}$ | $\frac{11}{21}$ | $\frac{11}{21}$ | $\frac{10}{21}$ | $\frac{1}{3}$ | $\frac{11}{21}$ | $\frac{11}{21}$ | $\frac{25}{42}$ | $\frac{25}{42}$ |

## B   Monte Carlo $\mathrm{PMI}_q$

*Proof of proposition 5.2.* Random contingency tables with fixed marginals can be generated in $\mathcal{O}(\min(N, k_A k_B \log N))$ [4, 26]. The error $a = \sqrt{\mathrm{PMI}_q(1 - \mathrm{PMI}_q)/N_{\text{samples}}}$ decreases with the inverse square root of the number of Monte Carlo samples $N_{\text{samples}}$. Hence we need $N_{\text{samples}} \leq a^{-2}/4$ samples to reach the approximation error $a$. $\square$

# C   The standardized Rand Index

For the proof of the runtime complexity of the $\mathrm{SMI}_2$, we use the fact that it is equivalent to the SRI (Corollary 2 in [30])

$$\mathrm{SRI} := \frac{\mathrm{RI} - \mathbb{E}[\mathrm{RI}]}{\sqrt{\mathrm{Var}[\mathrm{RI}]}} = \frac{\sum_{ij} \binom{n_{ij}}{2} - \mathbb{E}\left[\sum_{ij} \binom{n_{ij}}{2}\right]}{\mathbb{E}\left[\left(\sum_{ij} \binom{n_{ij}}{2}\right)^2\right] - \mathbb{E}\left[\sum_{ij} \binom{n_{ij}}{2}\right]^2} = \mathrm{SMI}_2 \,. \tag{17}$$

*Proof of Proposition 5.1.* Hubert and Arabie [11] derived the expected value under random permutation as

$$\mathbb{E}\left[\sum_{ij} \binom{n_{ij}}{2}\right] = \sum_{ij} \frac{\binom{a_i}{2}\binom{b_j}{2}}{\binom{N}{2}^2} \,. \tag{18}$$

For the variance, we also need the second moment. While Romano et al. [30] give a general formula in Eq. (13), it is impractical in the given form as the authors themselves note the runtime complexity is $\mathcal{O}(N^3 k_A \max(k_A, k_B))$. We observe that in the special case of the SRI, the higher moments can be simplified by leveraging the identity [39]

$$\mathbb{E}\left[\binom{n_{ij}}{2}^m \,\middle|\, n_{ij} \sim \mathrm{Hyp}(a_i, b_j, N)\right] = \sum_{l=2}^{\min(N, 2m)} \frac{S_{2,2}(m, l)}{2^m} \prod_{p=0}^{l-1} \frac{(a_i - p)(b_j - p)}{N - p}, \tag{19}$$

with the generalized Stirling Number $S_{2,2}(m, k)$ [22] and the hypergeometric distribution Hyp. Note that the right-hand side is completely independent of $n_{ij}$. Hence the expected values under the hypergeometric distributions in Eq. (13) in [30] can be calculated in $\mathcal{O}(\max\{k_A, k_B\})$ time, giving

$$
\begin{aligned}
\mathbb{E}\left[\sum_{ij} \binom{n_{ij}}{2}^2\right] = \Bigg( & 2\gamma_a \sum_{j=1}^{k_B} (N - b_j)(N - 3(b_j - 1))(b_j - 1)b_j \\
& + \sum_{i=1}^{k_A} a_i^2 (a_i - 1) \sum_{j=1}^{k_B} (4N - 5b_j + 3)(b_j - 2)(b_j - 1)b_j \\
& + \sum_{i=1}^{k_A} a_i^3 (a_i - 1) \sum_{j=1}^{k_B} (b_j - 3)(b_j - 2)(b_j - 1)b_j \\
& + (\gamma_a^2 - \gamma_{a,2}) \sum_{j=1}^{k_B} b_j (b_j - 1)(b_j - 2)(b_j - 3) \\
& + (\gamma_b^2 - \gamma_{b,2}) \sum_{i=1}^{k_A} a_i (a_i - 1)(a_i - 2)(a_i - 3) \\
& + (\gamma_a^2 - \gamma_{a,2})(\gamma_b^2 - \gamma_{b,2}) \Bigg) \\
& \frac{1}{4 \left(N(N-1)(N-2)(N-3)\right)},
\end{aligned}
\tag{20}
$$

for the most general case $N \geq 4$ and no cluster larger than $N - 2$, with

$$\gamma_a = \sum_{i=1}^{N} a_i(a_i - 1) \quad \gamma_{a,2} = \sum_{i=1}^{N} a_i^2(a_i - 1)^2 \quad \gamma_b = \sum_{j=1}^{N} b_j(b_j - 1) \quad \gamma_{b,2} = \sum_{j=1}^{N} b_j^2(b_j - 1)^2 \,. \tag{21}$$

The other cases can be treated analogously using Eq. (19). Hence the dominant factor for the runtime complexity of the SRI is the RI itself with $\mathcal{O}(k_A k_B)$ for the sum over all contingency matrix elements. □

## D Community detection parameter configurations

For the experiment in Figure 4c, we compared five community detection algorithms implemented in `networkit` [32]. Label Propagation and Degree Ordered Label Propagation do not have any parameters. The parameter choices for the other algorithms is listed in Table 4.

Table 4: Parameter configurations for the experiment on the email EU core dataset. For Leiden, we used all combinations of $\gamma$ and randomize.

| Algorithm | Parameters |
|---|---|
| Louvain | $\gamma \in \{0.001, 0.01, 0.1, 1.0\}$ |
| Louvain Map Equation | hierarchical $\in \{\text{True}, \text{False}\}$ |
| Leiden | $\gamma \in \{1 \times 10^{-6}, 1 \times 10^{-5}, 0.0001, 0.001, 0.01, 0.1, 1.0\}$ |
| | randomize $\in \{\text{True}, \text{False}\}$ |

## E Spectral clustering on image datasets

We conducted two additional experiments similar to the ones in Section 6.1, but using spectral clustering instead of $k$-means. We apply spectral clustering with varying number of clusters $k$ to the UCI image segmentation dataset [6] and a texture classification dataset from OpenML [36]. We compare each clustering solution with the ground truth labels and select the clustering with the highest $\text{RI}, \text{AMI}_2$ and $\text{PMI}_2$. Figure 5 shows the selection frequency of each number of clusters $k_{\text{selected}}$ after 1000 trials with different random seeds. In the case of the image segmentation dataset, different subsets of 1000 samples were chosen for each trial, due to the steep runtime requirements of spectral clustering. The results align with the $k$-means experiment in Section 6.1: The $\text{PMI}_2$ selects clusterings in a less biased way, in the sense that the selected number of clusters $k_{\text{selected}}$ is closer to the true number of clusters $k_{\text{true}}$ than for RI and $\text{AMI}_2$.

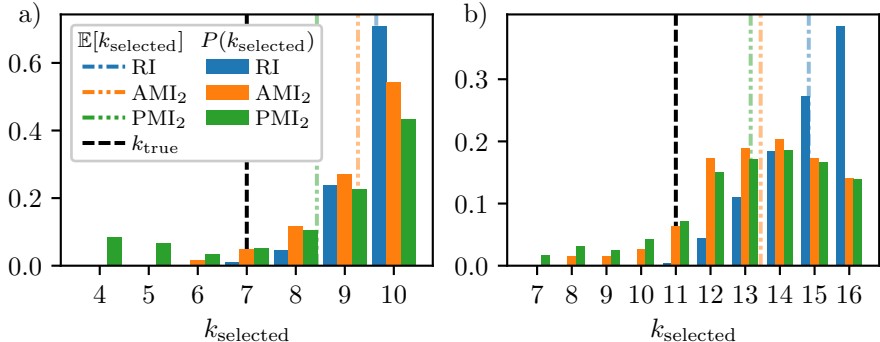

Figure 5: We apply spectral clustering to a) the UCI image segmentation dataset and b) a texture classification dataset. The number of clusters parameter $k$ is set to eleven values between $k_{\text{true}}/2$ and $3k_{\text{true}}/2$. We compare the resulting clusterings with the ground truth via RI, $\text{AMI}_2$, and $\text{PMI}_2$ and select the best clustering $k_{\text{selected}}$ according to each metric. We repeat the experiment with 1000 different random seeds and plot the selection probabilities of $k_{\text{selected}}$ for each metric. The $\text{PMI}_2$ selects candidates where the number of clusters is closer to the true number of clusters $k_{\text{true}}$ on average (dashed lines) compared to the RI and $\text{AMI}_2$.

