# OpenReview forum: "$p$-value Adjustment for Monotonous, Unbiased, and Fast Clustering Comparison"
_NeurIPS.cc/2023/Conference — NeurIPS 2023 poster_

### Official Review · Reviewer_wdxj · 2023-07-05

**Soundness:** 3 good
**Presentation:** 3 good
**Contribution:** 2 fair
**Rating:** 6
**Confidence:** 4

**Summary:**

p-value Adjustment. The p-value adjusted Rand Index is unbiased. The authors claim that its approximations outperform STD Mutual Information.
First, generalized MI relies on the TSALLIS entropy (same family of RENYI). AMIq comes from subtracting the expectation under random permutations. PMIq is derived and it is monotonic for q>=2 where q=2 is the Rand Index. Monotonicity is due to the properties of the bypass TSALLIS entropy.

**Strengths:**

This is a general tool for testing clustering algorithms. The properties of bypass entropies such as Tsallis are leveraged.
Computational complexity is quadratic.

**Weaknesses:**

It is quite theoretical and general. Poor experiments. Seems more a statistical paper. Please, motivate the use of bypass entropy estimators. It is more than a matter of efficiency. Other formal properties are also key (e.g. monotonicity).

**Questions:**

How critical is the choice of the Tsallis entropy? What about the  Kozackenko-Leonenko/Kraskow et al.’s approach? In ML there is an intererest in Rényi entropies and KNN graphs. Can you say anything about the properties of the graph associated with the clustering table.

**Limitations:**

The paper is general but few baselines are explored. Should be nice to test with segmented images or outputs of spectral clustering where there is an implicit bias. In other words, in my opinion the paper needs to tackle more recent datasets used in deep learning and image analysis to approach this result to the NEURIPS community.

---

> ### Author Rebuttal · Authors · 2023-08-07
>
> Thank you for your thoughtful comments and constructive feedback. We have taken all comments into consideration and summarize our response as follows:
>
> 1. **Experiments with segmented images or spectral clustering**
>
> 	We conducted two additional experiments using spectral clustering, on an image segmentation dataset [5] and a texture classification dataset [6]. The results are presented in Figure 1 in the attached PDF and confirm the experiments in Figure 4 in the manuscript. As the theoretical aspects of the $\\operatorname{PMI}\_2$ are the main focus of this work, we prioritize having the comparison to other clustering comparison metrics in the main body (See Reviewer heZh, PDF). The experiments on spectral clustering will be included in the appendix, due to the similarity with Figure 4 and the page limit.
>
> ---
>
> 2. **Choice of Tsallis entropy over Rényi entropy**
>
>     The Tsallis and Rényi entropy are distinct mathematical concepts that both generalize the Shannon entropy.  The Tsallis entropy replaces the logarithm with a modified $q$-logarithm (See Definition 2.1). This $q$-logarithm creates a link between information-theoretic and pair-counting measures [1]. For example, the adjusted Tsallis mutual information for $q=2$ is identical to the adjusted Rand Index. The Rényi entropy retains the conventional logarithm, such that this link to pair counting measures is not possible.  We rephrased lines 29f. and 86f. to further emphasize this point and refer to [1] for an in-depth discussion.
>
> ---
>
> 3. **Mutual information estimation (Kozachenko-Leonenko/Kraskow et al., nearest-neighbor graphs)**
>
> 	We assume you are referring to approaches to estimate mutual information of two *continuous* random variables, given a finite number of samples [2,3]. In this work, we focus on evaluating clusterings that consist of a *finite* number of clusters. Hence the cluster assignment can be understood as *discrete* random variable. Therefore the approach in [2,3] and KNN graph approaches like [4] are not applicable.
>
> ---
>
> 4. **Properties of the graph associated with the clustering table**
>
> 	We assume you refer to the contingency table of two clusterings $A$ and $B$ (Table 1). Note that the contingency table is not necessarily a square matrix, such that it can be seen as a block in the adjacency matrix of a bipartite graph at best. The nodes on one side of this bipartite graph would represent the clusters in clustering $A$ and on the other clusters in clustering $B$. Edges would represent intersections between the clusters with the edge weight equal to the size of the intersection. While this is an interesting perspective on contingency tables, it is unclear to us how this contributes to the main content of our work, the $\\operatorname{PMI}\_2$.
>
>
> [1] Romano, S., Vinh, N. X., Bailey, J., & Verspoor, K. (2016). Adjusting for chance clustering comparison measures. _The Journal of Machine Learning Research_, _17_(1), 4635-4666.
> [2] Kozachenko, L. F., & Leonenko, N. N. (1987). Sample estimate of the entropy of a random vector. _Problemy Peredachi Informatsii_, _23_(2), 9-16.
> [3] Kraskov, A., Stögbauer, H., & Grassberger, P. (2004). Estimating mutual information. _Physical review E_, _69_(6), 066138.
> [4] Pál, D., Póczos, B., & Szepesvári, C. (2010). Estimation of Rényi entropy and mutual information based on generalized nearest-neighbor graphs. _Advances in Neural Information Processing Systems_, _23_.
> [5] Image Segmentation. (1990). UCI Machine Learning Repository. https://doi.org/10.24432/C5GP4N.
> [6] Brodatz, P. (1966). Textures: A Photographic Album for Artists and Designers. *Dover Publications,Inc.*, New York.

---

> > ### Comment · Reviewer_wdxj · 2023-08-18
> > **Response to rebuttal**
> >
> > Thank you for doing the additional experiments and to answer the questions. I upgrade my rating.
> > Thanks a lot.

---

### Official Review · Reviewer_i4ZB · 2023-07-06

**Soundness:** 3 good
**Presentation:** 2 fair
**Contribution:** 3 good
**Rating:** 5
**Confidence:** 1

**Summary:**

The article introduces a new measurement called $\text{PMI}_q$ for comparing clustering methods. The name "p-value adjustment" comes from $\text{PMI}_1$, which represents the p-value of the mutual information's variation. This new metric has desirable properties, including type II unbiasedness and monotonicity. The paper claims that $\text{PMI}_q, q\ge 2$ is the first clustering comparison method to satisfy both of these properties.

The definition of $\text{PMI}_q$ involves taking an expectation over all permutations in $S_N$, which can be computationally challenging. To address this challenge, the author proposes a Monte Carlo estimation method for $\text{PMI}_2$ and applies it to empirical experiments.

The paper's contribution lies in proposing a new unbiased and provably monotonic method for addressing the existing bias problem in current clustering comparison metrics. The author provides detailed and sufficient proofs to support the proposed method.

The paper's limitations include its disorganized structure and lack of clarity in some sections. The article's figures and tables need improvement, and the author does not explicitly address any limitations or potential future research directions.

**Strengths:**

1. This paper provides valuable ideas and techniques for improving cluster comparison, particularly a new unbiased and provably monotonic method for addressing the existing bias problem, which may be useful to practitioners and researchers in the field of cluster comparison.

2.Detailed and sufficient proofs give relatively clear theoretical and technical support on new methods.

**Weaknesses:**

1. The article's structure is somewhat disorganized and can be challenging to read. To improve clarity, the authors could provide more examples and intuitive explanations for Definitions 3.1 to 3.4 and 4.1 to 4.4.

2. It would be beneficial to compare the proposed clustering comparison metric with other commonly used metrics in the field. By doing so, the authors could inform readers about whether these metrics possess desirable properties such as type II unbiasedness and monotonicity.

3. The figures and tables in the article need improvement, as some elements are unclear or difficult to read. For example, Figure 3a has too a large legend.

4. The results in Figure 2(c) could be more effectively presented to allow for easier differentiation across different N values. The authors may want to explore alternative visual representations or labeling techniques to better convey this information.

**Questions:**

1. Is $\text{PMI}_q$ type I unbiased?

2. Section 6 is unclear on how to calculate $k_{pred}$. Although $\text{PMI}$ can compare two clusterings, it is unclear which clusterings are being compared in the experiments discussed in Section 6. As such, further clarification is needed to understand how $k_{pred}$ is computed in this section.

**Limitations:**

1. The paper does not explicitly address any limitations or potential avenues for future research.

2. While the author asserts that $\text{PMI}_2$ is the first clustering comparison method to satisfy both type II unbiasedness and monotonicity, it is challenging to evaluate the significance of this claim without further comparison.

---

> ### Author Rebuttal · Authors · 2023-08-07
>
> Thank you for your thoughtful comments and constructive feedback. We have taken all comments into consideration and summarize our response as follows:
>
> 1. **Is $\\operatorname{PMI}\_q$ type I unbiased?**
>
>     Yes. Type I unbiased means that when you compare a fixed clustering $A$ to all permutations of any clustering $B$, the average metric value is the same for all $A$. The $p$-value of the mutual information ($\\operatorname{PMI}\_q$) tells you what percentage of all permutations of the data points would have led to higher mutual information. Now, if we average that percentage over all permutations we get a constant 50%. Formally we use Eq. (15) from Appendix A with with $A,B,\\tilde{B},\\mathcal{B}$ as in Appendix A to get $\\mathbb{E}\_\\mathit{B\\sim\\mathcal{B}}[\\operatorname{PMI}\_q(A,B)]=1-\\mathbb{E}\_\\mathit{\\tilde{B}\\sim\\mathcal{B}}[\\operatorname{PMI}\_q(A,\\tilde{B})]=1/2$. We added the proposition to the main body and a short proof to the appendix.
>
> ---
>
> 2. **What is $k\_\\text{pred}$ in Section 6/Figure 4?**
>
> 	Given a dataset with $k\_\\text{true}=10$ ground-truth clusters, we apply $k$-means clustering for different $k\_\\text{pred}\\in \\{10, 11, 12, 13, 14, 15 \\}$. We compare the $k$-means results with the ground truth and get for example $\\operatorname{PMI}\_2$ values of $[0.1, 0.3, 0.4, 0.2, 0.3, 0.1]$. In this case $k\_\\text{pred}=12$ had the highest score of $0.4$ and is selected. We repeat this experiment multiple times with different initializations of $k$-means and plot how often each $k\_\\text{pred}$ was selected in Figure 4a. To stress this selection step, we changed the axis labels to $k\_\\text{selected}$. We rephrased the Figure caption and extended the explanation of the experimental setup with examples in the main text.
>
> ---
>
> 3. **The paper does not explicitly address limitations or future directions**
>
> 	A limitation of the $\\operatorname{PMI}$ is its computational complexity. We provide an efficient standardized approximation to address this limitation. However, strictly speaking, the standardized approximation loses the theoretical guarantees of the $\\operatorname{PMI}$. We mitigate this limitation by also introducing a Monte Carlo approximation that retains the theoretical guarantees up to a tunable approximation error at the cost of higher runtime. Whether a weaker formulation of monotonicity can be found that carries over to the standardized approximation is an exciting direction for future research. Another open question is whether other metrics like the Jaccard Index could benefit from $p$-value adjustment and how it affects their monotonicity.
> 	We changed the conclusion to state those limitations explicitly. We also added the discussion about future avenues of research.
>
> ---
>
> 4. **Further comparison to other methods**
>
> 	We added a table comparing a total of 19 clustering comparison metrics. See comment to Reviewer heZh and PDF attached.
>
> ---
>
> 5. **Clarity and figures**
>
> 	We added intuitions for Definition 3.3, 3.2 and 3.4 (See Reviewer comment dKDt and the intuition for the PMI and type I bias in 1. of this comment). For a better understanding of Definitions 4.1 and 4.2 we added a reference to Definition 4 and Theorem 2 in [1], to stay within the page limit. We adjusted the legend size in Figure 3a and reworked the caption of Figure 4 (See 2. this comment).  We changed the presentation of Figure 2c, see PDF.
>
>
> [1] Gösgens, M. M., Tikhonov, A., & Prokhorenkova, L. (2021, July). Systematic analysis of cluster similarity indices: How to validate validation measures. In _International Conference on Machine Learning_ (pp. 3799-3808). PMLR.

---

> > ### Comment · Reviewer_i4ZB · 2023-08-16
> >
> > Thank you for your reply. I appreciate your response, and it is clear and logical to me. However, as I am not an expert in this particular field, I will maintain the score of "borderline accept".

---

### Official Review · Reviewer_dKDt · 2023-07-14

**Soundness:** 3 good
**Presentation:** 3 good
**Contribution:** 3 good
**Rating:** 8
**Confidence:** 3

**Summary:**

The paper presents a performance measurement method for cluster analysis. The method can avoid Type II bias that exists in previous approaches. A tractable approximation is given. The proposed method demonstrates advantages in both synthetic and real-world data sets.

**Strengths:**

The work consists of solid theoretical contributions and satisfactory empirical results. The finding is an important step forward in clustering research.

**Weaknesses:**

More empirical results would make the work more convincing.

**Questions:**

* It is better to elaborate the importance of Type II unbiasedness. Specifically, explaining Definition 3.3 in common words help readers understand the concepts and your contribution.

* There could be a mistake in Proposition 5.2. In the current formula, a higher accuracy leads to lower complexity, while a zero accuracy gives infinity.

---

> ### Author Rebuttal · Authors · 2023-08-07
>
> Thank you for your thoughtful comments and constructive feedback. We have taken all comments into consideration and summarize our response as follows:
>
> 1. **Further explanation of Type II unbiasedness**
>
>     Type I bias means that certain cluster sizes receive higher metric values. Type II bias gives higher relative rank to certain cluster sizes, when multiple clusterings are compared with a ground truth.  We included this explanation to help illustrate Definition 3.3.
>
> ---
>
> 2. **Definition of accuracy**
>
> 	An accuracy $a=0.1$ means the error on the quantity is below $0.1$, a lower value of $a=0.01$ is thus more accurate. We eliminate the confusion by renaming "accuracy" to "approximation error".
>
>   ---
>
> 3. **More empirical results would make the work more convincing**
>
> 	We added further experiments using spectral clustering on an image segmentation and texture classification dataset see comment to Reviewer wdxj and attached PDF.

---

> > ### Comment · Reviewer_dKDt · 2023-08-12
> >
> > The author's responses are satisfactory.

---

### Official Review · Reviewer_heZh · 2023-07-20

**Soundness:** 3 good
**Presentation:** 2 fair
**Contribution:** 2 fair
**Rating:** 6
**Confidence:** 1

**Summary:**

The paper introduces a new method called the p-value adjusted Rand Index (PMI2) for comparing clustering and community detection algorithms. The paper highlights the limitations of existing metrics, such as the Rand Index and Adjusted Rand Index, which suffer from bias and non-monotonicity issues. The PMI2 method addresses these issues by providing a type II unbiased and provably monotonic metric that has fast approximations. The paper also provides experimental results on image and social network datasets to demonstrate the effectiveness of the PMI2 method.


**Strengths:**

Originality: The paper introduces a new clustering comparison metric, the p-value adjusted Rand Index (PMI2), which addresses the limitations of existing metrics. The PMI2 method is the first to be type II unbiased and provably monotonic, and it has fast approximations.

Quality: The paper provides a thorough analysis of the limitations of existing clustering comparison metrics and demonstrates the effectiveness of the PMI2 method through experiments on synthetic benchmarks, image, and social network datasets. The paper also provides theoretical proofs of the PMI2 method's properties.

Clarity: The paper is well-organized and clearly presents the motivation, background, methodology, and experimental results of the PMI2 method. The paper also provides detailed explanations of the theoretical proofs and the approximations used in the method.

Significance: The paper's contributions are significant as they provide a more reliable and accurate way to evaluate clustering and community detection algorithms. The PMI2 method's properties make it a valuable tool for practitioners to choose better algorithms for their datasets. The paper's theoretical proofs and experimental results also provide a deeper understanding of clustering comparison metrics and their limitations.

**Weaknesses:**

One potential weakness of the paper is that it does not compare the PMI2 method with other recently proposed clustering comparison metrics. While the paper provides a thorough analysis of the limitations of existing metrics, it would be valuable to compare the PMI2 method with other state-of-the-art methods to demonstrate its superiority.

Another weakness is that the paper does not provide a detailed explanation of the approximations used in the PMI2 method. While the paper mentions that the method has fast approximations, it would be helpful to provide more information on how these approximations work and how they affect the accuracy of the method.

**Questions:**

A more extensive comparison of the PMI2 method with other recently proposed clustering comparison metrics to demonstrate its superiority would help to further validate the effectiveness of the PMI2 method.

A more detailed and intuitive explanation of the approximations used in the PMI2 method and how they affect its accuracy would help readers to better understand the method and its limitations.

**Limitations:**

Yes

---

> ### Author Rebuttal · Authors · 2023-08-07
>
> Thank you for your thoughtful comments and constructive feedback. We have taken all comments into consideration and summarize our response as follows:
>
> 1. **Further comparison to other methods**
>
> 	We added a table comparing a total of 19 clustering comparison metrics (See attached PDF), mostly adapted from a systematic review of cluster comparison metrics [1]. We added a column for type II bias and the $\\operatorname{PMI}\_q$ for $q\\in\\{1,2\\}$. For all metrics except the $\\operatorname{PMI}\_q$, we provide examples that violate type II unbiasedness in the appendix.
>
> ---
>
> 2. **Details about approximations**
>
> 	**Standardized approximation (**$\\mathbf{q=2}$**):** Intuitively, we approximate the true (discrete) distribution of $\\operatorname{MI}\_q$ with a (continuous) normal distribution (See Figures 2a and b). We added this intuition to Section 5.1. As per Reviewer i4ZB's comment, we remodeled Figure 2c, to highlight the effect on accuracy (See attached PDF).
>
> 	**Monte Carlo approximation:** Given two clusterings $A$ and $B$, we sample contingency matrices (c.f. Table 1) with their cluster sizes $\\{a\_1, \\dots, a\_{k\_A}\\}, \\{b\_1, \\dots, b\_{k\_B}\\}$ using [2,3]. The fraction of matrices with $MI\_q$ lower than $MI\_q(A,B)$ then approximates the true $p$-value. We added this explanation to Section 5.2.
>
> [1] Gösgens, M. M., Tikhonov, A., & Prokhorenkova, L. (2021, July). Systematic analysis of cluster similarity indices: How to validate validation measures. In _International Conference on Machine Learning_ (pp. 3799-3808). PMLR.
> [2] Boyett, J. M. (1979). Algorithm as 144: Random r× c tables with given row and column totals. _Journal of the Royal Statistical Society. Series C (Applied Statistics)_, _28_(3), 329-332.
> [3] Patefield, W. M. (1981). Algorithm AS 159: an efficient method of generating random R× C tables with given row and column totals. _Journal of the Royal Statistical Society. Series C (Applied Statistics)_, _30_(1), 91-97.

---

### Official Review · Reviewer_x3cb · 2023-07-26

**Soundness:** 3 good
**Presentation:** 3 good
**Contribution:** 2 fair
**Rating:** 6
**Confidence:** 2

**Summary:**

The paper proposes an improved clustering comparison metric. While many current metrics fix a "type I" bias (being biased towards certain cluster size distributions), many still suffer from a "type II" bias (a bias towards certain clusterings when they are compared to a ground truth clustering). Building on top of the Adjusted Rand Index, the type II bias is analyzed and a fast  and a p-value adjusted variant is introduced which is type II unbiased. A previous approach also solves this issue (SMI), however, it suffers from being non-monotonous, while the proposed metric is monotonous, i.e. when modifying a clustering in such a way that it objectively becomes better, the metric reflects this. While the proposed PMI metric relies on Monte Carlo approximation and thus is fairly costly, the SMI_2 metric is fast to compute and with a small modification seems to be a good approximation to the PMI_2. Results on three different datasets show that the best selected clusterings according to the PMI metric seem more reasonable when compared to two baselines.

**Strengths:**

- The paper is fairly easy to follow and seems to be well written. Even though I am not an expert in this field, I appreciate that the paper guided me through the most important considerations and different steps to get to the proposed metric.
- Monotonicity is a reasonable aspect to consider from a clustering comparison metric, as also highlighted by Gösgens et al. [9]. Properly defining this concept and showing that the proposed PMI_q metric is monotonous for q>=2 is a valuable contribution.
- The fast approximation of PMI_2 makes the proposed metric more practically applicable.
- Code is provided.

**Weaknesses:**

My main concern is the overall practical significance of the proposed metric. While SMI is not monotonous according to Gösgens et al. [9], there seems to be empirical evidence for SMI_q. A proper proof for PMI_q is very nice, but the real-world experiments are based on the SMI_2 based approximation of PMI_2. In my mind the obvious baseline to compare to, would be SMI_2, but this is omitted for some reason and I really wonder why. Even though PMI_2 seems to select the more reasonable clusterings in all experiments, the margin to AMI_2 is not very large in two of the three setups and I would not be surprised if it is even smaller for the SMI_2 score. Adding such a comparison would give a more complete picture of how the different metrics compare.

**Questions:**

Given that you only theoretically prove that PMI_2 is type II unbiased, but you then in practice approximate it with SMI_2, is there some guarantee that this property still holds?

**Limitations:**

Limitations or potential negative impact have not been discussed in the paper. I would not be able to come up with a direct negative societal impact myself, but a discussion of potential limitations of the approach (if any are clear) would have been valuable.

---

> ### Author Rebuttal · Authors · 2023-08-07
>
> Thank you for your thoughtful comments and constructive feedback. We have taken all comments into consideration and summarize our response as follows:
>
> 1. **Concern about practical significance**
>
> 	- While technically the $\\operatorname{SMI}\_2$ was introduced in [1], the authors only provide the runtime complexity $\\mathcal{O}(N^3 k\_A \\max\\{k\_A,k\_B\\})$ of the $\\operatorname{SMI}\_q$ for general $q$, which is intractable in many real situations. Therefore, to the best of our knowledge, the $\\operatorname{SMI}\_2$ has not seen any adoption in practice. We reformulate the $\\operatorname{SMI}\_2$ and provide an implementation with a runtime of $\\mathcal{O}(k\_A k\_B)$, thus opening the $\\operatorname{SMI}\_2$ for practical applications. Further, we normalize the $\\operatorname{SMI}\_q$ in such a way, that it is closely related to the $\\operatorname{PMI}\_q$. The latter is monotonous for $q=2$ but not $q=1$, providing further arguments for practitioners to chose the $\\operatorname{SMI}\_2$ over the $\\operatorname{SMI}\_1$.
>
> 	- Ultimately, the nice theoretical guarantees only hold for the $\\operatorname{PMI}\_2$ and not the $\\operatorname{SMI}\_2$. Therefore we also introduce an unbiased Monte Carlo estimator that retains these guarantees up to a tunable error bound at the cost of a higher runtime. Practitioners can chose our MC implementation if theoretical guarantees are required.
>
> 	- The difference between $\\operatorname{AMI}\_2$ and $\\operatorname{PMI}\_2$ in Figure 4 is subtle, but this is to be expected. Type II bias correction is just one step forward in clustering comparison and does not turn existing assessments based on metrics like the $\\operatorname{AMI}\_2$ on its head. We included this train of thought into the discussion in Section 6.
>
> ---
>
> 2. **Why we base our real-world experiments on $\\Phi(\\operatorname{SMI}\_2)$**
>
> 	While the differences between the $\\operatorname{PMI}\_2$ and the $\\operatorname{AMI}\_2$ in Figure 4 are already subtle, the difference between the two approximation schemes would be virtually non-existent (as you suspected). We think that for many applications, the practical benefits of the $\\Phi(\\operatorname{SMI}\_2)$ approximation outweigh the theoretical guarantees of a Monte Carlo approximation (See Section 5), which is why we chose it for the real-world experiments. As the $\\Phi(\\operatorname{SMI}\_2)$ is just a normalized $\\operatorname{SMI}\_2$, a comparison to the latter would not provide further insight. We reformulated the explanation in Section 5 and 6 to explain this choice better.
>
> ---
>
> 3. **Discussion about limitations**
>
> 	A limitation of the $\\operatorname{PMI}$ is its computational complexity. We provide an efficient standardized approximation to address this limitation. However, strictly speaking, the standardized approximation loses the theoretical guarantees of the $\\operatorname{PMI}$. We mitigate this limitation by also introducing a Monte Carlo approximation that retains the theoretical guarantees up to a tunable approximation error, at the cost of higher runtime. We explicitly added a discussion of these two limitations to the manuscript. (See also our comment to Reviewer i4ZB)
>
> [1] Romano, S., Vinh, N. X., Bailey, J., & Verspoor, K. (2016). Adjusting for chance clustering comparison measures. _The Journal of Machine Learning Research_, _17_(1), 4635-4666.

---

> > ### Comment · Reviewer_x3cb · 2023-08-16
> > **Post-rebuttal opinion**
> >
> > Thank you for the detailed answers to my and other reviewers questions. I do feel like with the additional provided results the paper will be stronger.
> > I certainly appreciate that the paper introduces an efficient SMI_2 implementation, nevertheless, I still think the initially submitted paper pushes the idea that PMI_2 is type II unbiased and that this is the real metric you are presenting. As you say yourself, the nice guarantees do not hold for SMI_2 and thus they don't hold for the efficient PMI_2 approximation either. In the end this boils down to the paper selling method A, stating the type II unbiasedness is important, but then evaluating method B, which is not type II unbiased. As such I find that the paper isn't really written in the clearest of ways and given that I can't really see your discussion of this in the revised paper, I find it a bit difficult to now increase my score.

---

> > > ### Author Response · Authors · 2023-08-17
> > >
> > > Thank you for your thoughtful feedback and for acknowledging the additional results we've provided to strengthen the paper.
> > >
> > > We present the $\\operatorname{PMI}\_2$ as an ideal method, with the limitation that it is computationally intractable. Therefore we provide two ways to trade exactness for speed.
> > >
> > > 1. **Why do we stress the $\\operatorname{PMI}\_2$?**
> > >
> > >     The $\\operatorname{AMI}\_2$ can be seen as a first-order correction, adjusting the $\\operatorname{MI}\_2$ by its first statistical moment. Standardization $\\operatorname{SMI}\_2$ (also known as Z-Score) corrects for the second statistical moment, and one might ask how to include the third, fourth, and higher moments. The $p$-value comprises all the information about the distribution. In that sense, the $\\operatorname{PMI}\_2$ is the ultimate goal of such a discussion incorporating all statistical moments (think Gram Charlier A Series).
> > >
> > >     More than that, we prove the $\\operatorname{PMI}\_2$ to be type II unbiased and monotonous. While computationally intractable, we provide two ways to trade exactness for speed.
> > >
> > > ---
> > >
> > > 2. **Trade-off between exactness and speed**
> > >
> > >     We stress the $\\operatorname{PMI}\_2$ because it allows for multiple, fundamentally different approximation approaches, two of which are presented in the paper:
> > >
> > >     - **Monte Carlo** preserves the theoretical properties up to a gaussian error $a$ (in the central limit theorem). The error can be reduced at the expense of a higher runtime. For errors < 0.001, for example, the MC approximation is in many cases faster than the $\\operatorname{SMI}\_1$ (See Figure 3a, we added a=0.001 to the legend). However, it is still inefficient compared to $\\Phi(\\operatorname{SMI}\_2)$.
> > >     - **$\\Phi(\\operatorname{SMI}\_2)$** can be understood as a second order Gram Charlier A expansion as outlined above. Higher orders could also be calculated using Eq. 19 in Appendix C, but we focus on the second order for simplicity. An exact error estimation is difficult and beyond the scope of this paper. Therefore we study the accuracy of this approach experimentally (Figure 2c) and show a comparison to the Monte Carlo trade-off in Figure 3b. For an empirical study of the type II unbiasedness of $\\operatorname{SMI}\_2$ see Figure 10 in [1]. We conclude that the MC approach should be used when theoretical guarantees are required and the $\\Phi(\\operatorname{SMI}\_2)$ should be used when the MC approach is too slow.
> > >
> > > Unfortunately, we cannot upload a revised version of the manuscript. However, we will include the discussion in 2. in Section 5.
> > >
> > > [1] Romano, S., Vinh, N. X., Bailey, J., & Verspoor, K. (2016). Adjusting for chance clustering comparison measures. The Journal of Machine Learning Research, 17(1), 4635-4666.

---

> > > > ### Comment · Reviewer_x3cb · 2023-08-18
> > > >
> > > > I'm mostly wondering about the practical implications of the proven properties of $\text{PMI}_2$. One impression I got from the paper was that fixing the type II unbiased is a very important property, but none of the experiments really seem to show this, since the approximations are doing just fine. So my question remains a bit, whether fixing the type II bias is the reason this metric works better, or if actually something else could be the reason. You comment that we should use the MC approach when theoretical guarantees are required, but actually demonstrating such a case would be significantly more convincing to me.
> > > >
> > > > Nevertheless, you managed to convince me a bit more and I am slightly raising my rating, but please make sure to discuss this in the final version of the paper. Of course the proof for $\text{PMI}_2$ has it's own value, but it should be very clear that this proof doesn't hold for the approximation, but that it empirically still seems to work better.
> > > >
> > > > Also, I do want to highlight to the ACs again that I am absolutely not an expert in this field and as such I was not able to fully check if the proof is actually correct or not. My rating should still be taken with a grain of salt.

---

> > > > > ### Author Response · Authors · 2023-08-18
> > > > >
> > > > > Thank you again for the nuanced feedback. We genuinely think that this discussion helped improve the paper. We will add this discussion to Section 5 and highlight how the approximations affect the theoretical guarantees, which aligns with the other reviewers' suggestions.

---

### Author Rebuttal · Authors · 2023-08-07

We sincerely appreciate all reviewers’ time and efforts in reviewing our paper. We thank you all for the insightful and constructive suggestions, which helped further polish our paper. We attached a PDF with three improvements to our paper that were stimulated by the reviewers' comments:

- We added a table comparing the proposed $\operatorname{PMI}_2$ to 18 other clustering comparison metrics from the literature.
- We conducted two more experiments using spectral clustering on an image segmentation and a texture classification dataset.
- We modified Figure 2c to differentiate different dataset sizes $N$ better visually.

We included numerous other clarifications, additions, and reformulations in the manuscript, which we summarize in the direct replies to the reviews.

---

### Decision · Program_Chairs · 2023-09-21

**Decision:**

Accept (poster)

**Comment:**

This is a borderline paper, but during the reviewer discussion and based on my own reading, I find that there are several nice ideas that are already mature in the current version, and room for further improvement with the additional experiments and promised discussions. Please carefully incorporate the constructive comments of the reviewers in preparing the revised version of the manuscript.